# Sustainability and Quality Management in the Italian Luxury Furniture Sector: A Circular Economy Perspective

**Marica Barbaritano, Laura Bravi \***  **and Elisabetta Savelli**

Department of Economics, Society, Politics, University of Urbino Carlo Bo, Via Saffi 42, 61029 Urbino, Italy;
marica.barbaritano@uniurb.it (M.B.); elisabetta.savelli@uniurb.it (E.S.)
**\*** Correspondence: laura.bravi@uniurb.it; Tel.: +39-0722-305571

**Abstract:** The growing attention paid to global environmental risks has gradually raised interest, both on the agendas of firms and governments towards the development of new business models such as Circular Economy. This study is focused on the luxury furniture industry and it is aimed at investigating how much furniture companies know about Circular Economy practices, what they specifically do for implementing them and what factors motivate, support or hinder their adoption. The role of product and process certifications in developing such sustainable practices is also analyzed, given their importance for implementing environmentally sustainable practices. The research method is based on a qualitative multiple case study carried out on four Italian companies operating in the luxury furniture industry. A worthy degree of awareness and knowledge of Circular Economy principles emerged from the analysis. Nevertheless, furniture companies analyzed are still little involved in Circular Economy practices, especially concerning reuse and recycle actions, which are particularly important within this perspective. Similarly, very little use of process and product certifications emerged from the study. Therefore, a potential gap seems to arise between the positive attitude towards Circular Economy practices and their actual implementation, which suggests useful implications for both institutions and managers involved in sustainable development processes.

**Keywords:** sustainability; circular economy; quality management; luxury furniture; Italy

## 1. Introduction

In recent decades, growing attention has been paid to global environmental risks and related consequences, such as the increasing amount of $CO_2$ emissions, global warming, deforestation, acid rains and the depletion of resources, which are threatening humanity's survival [1,2]. Since the early 2000s, more efficient resource allocation and utilization are expected to increase overall company competitiveness as well as improve the well-being of society and reduce environmental damage and economic inequalities [3,4].

In this context, the concept of Circular Economy has gradually raised and received growing importance both on the agendas of firms and governments [5]. It has been defined as "an industrial economy that is restorative or regenerative by intention and design" [6]. Thus, Circular Economy represents a new business model, useful to achieve sustainable development [7–13], as well as a fair society, based on early childhood development interventions, increasing equality in the workplace, preventive health care and health behavior campaigns, and a better distribution of monetary income [14].

Economic and financial resources are needed to develop such a business model [15], while effective information systems and proper skills are required for planning and managing resource reduction, reuse and recycling [8]. Overall, companies and, more general, industries, could be forced

to reorganize their industrial value chains in a way that a sustainable use of resource and treatment of waste can be effectively implemented within the production process. Notwithstanding these changes, which could sometimes limit the development of Circular Economy among firms, its practical implementation will have positive effects on the economic dimension as well as on environmental and social ones [7]. Based on Circular Economy principles, companies can convert waste streams into income ones, using proper infrastructure for waste treatment [16]. Particularly, Circular Economy can help companies to reduce costs, thanks to an overall increase in the re-use of materials, and to enhance their competitiveness, mainly due to a decrease in the consumption of raw materials and the effects related to their price volatility [17]. As noted by Rizos and colleagues [17], circular business models can also lead to technological and organizational innovation, in addition to new employment opportunities, thus improving the overall well-being of society [18].

These opportunities could be particularly important in specific contexts, such as luxury, which includes companies producing manifold items (from clothes to furniture, to cars) characterized by "elitism, distinction and status, rarity, reputation, creativity, power of the brand, hedonism and refinement" [19]. The term "luxury", in this study, will be used for indicating products (specifically, furniture ones) that are implicitly designed to please and satisfy particular individuals' needs. In this regard, the luxury connotation of goods is not based purely on accessibility, as it is mainly embedded in the individual's desire for it [20].

Recent studies show that few luxury companies still take sustainable development positions, notwithstanding the increasing attention towards sustainable luxury goods on the part of the customers [21]. As stated by Kapferer and Michaut-Denizeau [21], luxury companies often use rare raw materials (e.g., animal skins, gold, or gemstones); promote animal treatments (e.g., the exploitation of crocodile farms, the killing of baby seals for fur); adopt manufacturing methods (e.g., mercury for tanning skins) that pollute the local environment, or destroy the environment itself (e.g., the use of endangered tree species and exploitation of rare water resources by luxury hotels located in poor countries). In this scenario, the role of some luxury clothes brands (e.g., Burberry, Chanel, Luis Vuitton) and retail giants (e.g., H & M) cannot be ignored, as they have admitted to burning unsold stock, thus becoming bad examples of wasteful and non-sustainable actors (www.newsweek.com).

Within this context, the present study will focus attention on luxury furniture companies, which offer products made out of various materials such as wood, metal, glass and leather, made from high-quality resources and reputed manufacturing processes. Notably, luxury furniture includes "movable pieces, which showcase the best of an elite quality and design associated with a certain era" [22]. To our knowledge, newspapers have not reported similar cases of the bad management of "elite" furniture stocks, such as the aforementioned ones. On the contrary, research into the web pages of some of the major Italian companies contacted for this study found the words "sustainability" and "eco-design" often combined with the name of the brand, thus underlining the importance they give to sustainability and their overall attention towards good sustainable practices.

Overall, the furniture sector accounts for a considerable portion of global trade. According to the latest report by Zion Market Research [23], the global furniture market was valued at approximately USD 331.21 billion in 2017 and it is expected to reach approximately USD 472.30 billion by 2024, growing at a compound annual growth rate (CAGR) of approximately 5.2% between 2018 and 2024. Approximately one-quarter of the world's furniture is manufactured within the European Union, representing an €84 billion market that equates to a EU28 consumption of about 10.5 million tons of furniture per year, while employing approximately 1 million workers [23]. Similarly, the luxury furniture business is an area with a great deal of growth. It was valued at approximately USD 23.05 billion in 2017 and it is expected to generate revenue of approximately USD 30.28 billion by the end of 2023, growing at a CAGR of approximately 4.65% between 2017 and 2023. Notably, Europe dominates the global market, with revenue estimated at USD 7329.8 million, due to the increasing disposable incomes of consumers and rising economic growth [22].

The furniture sector appears to be particularly important from environmental standpoints since it is characterized by an intensive use of virgin raw materials and because the large use of adhesives, dyes and coating materials in furniture production results both in the emission of large volumes of volatile organic compounds and waste production [24]. As concerning the waste issue, in particular, in 2017, the EU28 total amount of furniture waste equated to 10.78 million tons annually, accounting for over 4% of the total municipal solid waste (MSW) stream. Waste arising from commercial sources contributes 18% of total furniture waste generation. Moreover, 80 to 90% of EU furniture waste in MSW is incinerated or sent to landfills [25].

With respect to luxury furniture, the great use of raw materials and energy is generally recognized. Raw materials mainly include wood, which accounts for the largest market share, followed by metal and other popular resources such as glass, leather and plastic [22]. All these materials require different amounts of embodied energy. For instance, natural materials, such as stone and wood, imply less embodied energy than steel and plastic [26]. This suggests that the definition of an overall model of resources used for this industry is rather difficult. However, the large employment of natural resources also suggests the importance of moving towards sustainable practices for reducing both the environmental damage and social impact of firms' production [27].

Hence, the importance of Circular Economy in this context looms, as it enables the prevention of resource depletion, the maintenance of materials and products in cycles, and the recycling of potential waste [6]. Besides the environmental advantages, the adoption of circular practices in the luxury furniture sector can also improve financial performances. For example, companies engaging in Circular Economy, instead of spending money on waste management, could gain from selling waste materials to a company that uses them as input or find ways to reuse or recycle the waste themselves [28].

Nevertheless, to the best of our knowledge, there is a lack of contributions examining the concept of Circular Economy among furniture companies, especially those operating in the luxury market segment [21]. Extant literature investigated the role of eco-design [29,30], the increasing use of recycling raw materials [31] and the growing adoption of renewable energy by furniture companies [27,32]. Certainly, these topics are strictly related to the concept of Circular Economy, but little is known about other practices, such as the recovery/reconversion of waste materials to create new products on which Circular Economy lays its foundations [33].

Therefore, the main objective of this study is to explore to what extent luxury furniture companies are aware of Circular Economy principles, how they are implemented within them and which factors can influence their adoption. In this way, the research contributes to address a gap, still recognized in the literature, concerning the overall lack of knowledge on the concept of Circular Economy and characteristics, since it is yet in its early stage of development [28].

At an operational level, quality management practices are also investigated in this study, since they can be particularly helpful for managers in implementing environmentally sustainable practices, which, in turn, are critical within a circular business context [34]. Quality management practices, including Quality, Environmental and Corporate Social Responsibility Management Systems (QMS–EMS–CSRMS) and Product Certifications (PC) are investigated to understand how much they are adopted by luxury furniture companies and how they contribute to support their overall approach towards sustainability and circularity.

Two main research questions operationalize the above purposes:

(RQ1) How much do luxury furniture companies know about Circular Economy? What do companies do specifically? What factors motivate, support or hinder the adoption of Circular Economy practices?

(RQ2) Do luxury furniture companies adopt product and process certifications? For what reasons and purposes?

The study is exploratory in nature, since the Circular Economy model is still recent and the process of transitioning from the linear economic model is in its initial phase, as well as the scientific research

regarding the luxury furniture industry. Accordingly, the research is based on a qualitative multiple case study carried out on four luxury companies operating in the Italian furniture sector. The findings add further knowledge to the scientific debate on Circular Economy, specifically concerning the furniture sector. Moreover, practical implications for companies derive from the analysis of factors affecting the adoption of Circular Economy practices and product/process certifications, which are particularly useful for those who are moving towards circularity.

The rest of the paper is structured as follows. Section 2 provides a literature background about the concept of Circular Economy and the main process and product certifications that can be adopted under a quality management perspective. Section 3 describes the research method, while Section 4 illustrates the main research findings. Section 4 discusses the results, also in light of previous research, and provides theoretical and practical implications. The final section highlights the main limitations of the study and proposes directions for future research.

## 2. Theoretical Background

### 2.1. The Concept of Circular Economy

In recent years, Circular Economy has received growing attention worldwide [6,7,11,35–39]. Notably, a considerable amount of studies has been carried out by Chinese scholars [8,10,35,38,40,41], as their country is facing growing social and environmental problems due to its rapid and continuous economic development.

Based on Boulding's studies [42] and his idea of economy as a closed economic system, the concept of Circular Economy was first introduced by Pearce and Turner [43]. They argued that the traditional open-ended economic system based on the paradigm "take, make and dispose" paid scarce attention to environmental issues and, more specifically, to the recycling practices which can occur in both production and consumption processes [9]. Thus, they conceptualized a closed-loop of materials in order to promote harmony between the economic system and the ecosystem.

Following Pearce and Turner's conceptualization [43], Yuan and colleagues [33] stated that "the core of [Circular Economy] is the circular (closed) flow of materials and the use of raw materials and energy through multiple phases". More recently, Circular Economy has been defined by the Ellen MacArthur Foundation [6] as "an industrial economy that is restorative or regenerative by intention and design". Even today, this definition is recognized as the most renowned definition by several scholars [44], and this is the definition that we take into account in developing this work. It proposes Circular Economy as a new business model, useful to achieve sustainable development and a fair society [7–13]. This is a model that replaces the "end-of-life" concept with that of reducing alternative reuse, recycling and recovery of materials in production and consumption processes (Figure 1).

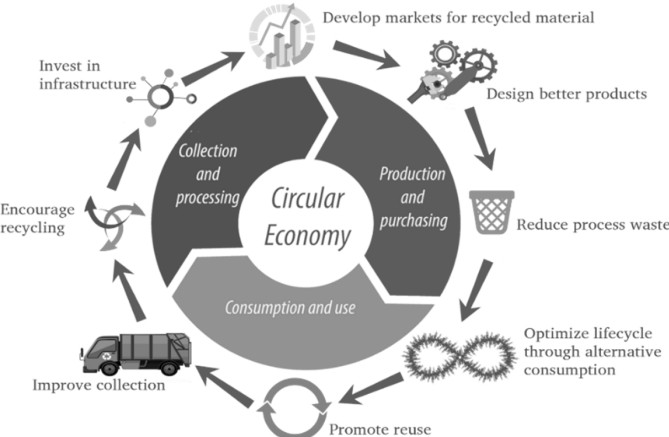

**Figure 1.** The Circular Economy model. Source: Sustainable Global Resources Ltd.—Recycling Council of Ontario.

This new business model can lead to a reduction of dependence on natural resources and, consequently, it results in a lower exposure to the negative effects of resource price shocks with an increase in company competitiveness [6,37].

Similar to what was expressed by the concept of sustainability [45], Circular Economy is also based on a global perspective, stressing a compelling need of better integration between economic, social and environmental aspects, in order to achieve different expectations by both companies and countries. In other words, Circular Economy can be considered as a way to reinterpret the traditional economic paradigm with sustainability principles in mind [46].

There is a wide consensus that companies and policymakers are the main actors for the transition to a circular system. Thus, in recent decades, the concept of Circular Economy has received growing importance in governmental policies [5]. Germany enacted the first legislative act related to Circular Economy in 1996, namely "Closed Substance Cycle and Waste Management Act" [38], with the aim of achieving an environmental waste treatment through suitable management. Later, Japan and China incorporated Circular Economy concerns in their policies with two legislative acts, i.e., "Basic Law for Establishing a Recycling-Based Society" [47] and "Circular Economy Promotion Law of the People's Republic of China" [48], respectively. In Europe, Circular Economy has been recently promoted through the "Waste Directive 2008/98/EC" [49] and the "Circular Economy Package" [50].

## 2.2. Factors Affecting the Implementation of a Circular Economy Model

Zhu and Qiu [51] pointed out that implementing Circular Economy requires the compliance of three principles, namely Reduce, Reuse and Recycle (i.e., 3R principles), which have to be embedded in production and consumption processes. Reduce is aimed at minimizing the input of raw materials, energy and waste by increasing the efficiency in production and consumption processes, for example by introducing simplified packaging or more compact products and household appliances [7,38]. Reuse encompasses operations by which resources or components can be used again for the same purpose they were designed [52]. It implies that products are used to their maximum potential in order to extend their life [39]. Finally, Recycle relates to the possibility of reprocessing waste materials and components into production processes, thus resulting in a decrease of negative environmental consequences [53–55]. Notably, the compliance of this principle is not always possible depending on materials used in production processes and for the packaging. For instance, some types of plastic waste cannot be recycled, due to the presence of ink and metals [39], while metals can be recycled many times [56], according to their composition (pure or alloy), intrinsic characteristics and, at least, until the value recovered would be higher than the cost incurred in recycling them [57].

The Ellen Macarthur Foundation [6] integrated the above 3R principles by four additional ones. The first one emphasizes the relevance of design stage in order to limit waste discharge in landfills. By introducing a new classification of materials into "nutrients" and "technical", the second principle states that nutrient materials can be safely reintroduced to the biosphere, while technical materials have to be designed in a way to be reused. The third additional principle recognizes the importance of renewable energy to reduce energy dependence and, more generally, to improve the overall adaptability of the economic system. Finally, the concept of eco-design becomes another basic principle of the Circular Economy paradigm, as it can help companies in internalizing externalities related to design, by taking into account environmental impacts into product development practices [39]. Notably, eco-design implies the continuous search for innovative solutions that limit industrial waste and improve the use of cleaner materials in all stages of the value creation process [58]. It enhances both product and process innovation, by integrating sustainability and environmental aspects from the earliest stages of the product development [26,59]. Starting from raw materials, eco-design encourages the saving of resources by creating, for example, furnishing objects, the components of which are readily separable at the time of their disposal or by using raw materials that are easier to recycle than wood (e.g., aluminum and glass). In the production phases, eco-design recommends the increasing use of water paints in place of chemical ones, while during the assembly and finishing steps, eco-design

invites the use of machinery with high-energy efficiency as well as glues containing no toxic elements. Finally, regarding distribution, eco-design suggests optimizing the storage of goods, to assure better use of spaces and to reduce the number of trips [60].

Besides the above principles, the proper use of wood raw materials, the overall resources consumption and the recovery of wood and other material scraps can be considered as crucial issues in order to produce more environmentally friendly goods, particularly within the furniture sector, characterized by intensive wood use [61].

Several conditions are required for implementing the above Circular Economy principles. In detail, economic and financial resources are needed [15] for allowing the use of advanced technology, sustaining the energy consumption required to carry out recycle activities [62] and for promoting adequate waste disposal [63]. Geng and Doberstein [8] further stressed the role of an efficient information system for planning and managing all activities related to resource reduction, reuse and recycle. Finally, given the critical role of consumers in Circular Economy [48,64], the overall circular supply chains should also consider consumption processes in addition to production and distribution. Particularly, the degree of consumer awareness about environmental and sustainable issues could highly affect the practical implementation of a circular business model [65].

By considering the luxury setting, the adoption of Circular Economy practices and, more generally, of sustainability principles appears to be even more unusual and difficult for different reasons. As argued by Kapferer and Michaut-Denizeaut [21], luxury firms are usually focused on high quality and they are often family firms. Consequently, they tend to control the whole supply chain, from the raw materials to merchandising, and they are not usually interested in diffusing strategic information concerning themselves and their functions, as they are mainly aimed at maintaining "the dream image they are selling". Thus, also from a communication standpoint, sustainability values and practices become less relevant for building the whole image of the company. Moreover, luxury consumer purchases are so unusual, irregular, and expensive, that questions concerning manufacturing and firms' processes "are remote from buyers' decision criteria". Hence their attention towards sustainable behaviors of companies is very limited. In summary, the adoption of Circular Economy and sustainable practices by luxury companies could be quite difficult and reduced by communicative and structural characteristics of the industry as well as by the poor consumer awareness of such practices [19].

Therefore, the present study will focus not only on the analysis of how the concept of Circular Economy is perceived among luxury furniture firms but, especially, on factors enabling or preventing its implementation within this context.

*2.3. How to Develop Corporate Sustainability and Circularity: The Role of Quality Certifications*

Quality management (QM) may be particularly helpful to aid managers in implementing environmentally sustainable practices, thus moving towards a wider Circular Economy model [34]. Quality management and environmental sustainability have a long-run organizational view and focus on both economic and social well-being. To reach the maximum benefits from both QM and environmental sustainability, managers should adopt an integrated, multifunctional, and organizational approach involving the whole value chain [66].

Quality management tools such as Integrated Management Systems (IMSs), which include the development of Quality, Environmental and Corporate Social Responsibility Management Systems (QMS–EMS–CSRMSs) and Product Certifications (PCs) are particularly useful to help companies in developing environmentally sustainable practices [67].

In the early stages of introduction, IMSs were particularly relevant in highly demanding activity sectors, like the automotive and aeronautical ones [68]. Later, they gradually spread across all industries given the increasing importance of managing economic as well as environmental and social outcomes of any business around the world [69,70].

IMSs combine all components of a business into one coherent system, which allows organizations to achieve their purposes and mission [71] through a proper management of processes or activities that

satisfies the stakeholders' objectives, while meeting health, safety, environmental, security, ethical or any other identified requirement [72].

Several International and European recognized standards can be adopted for developing the IMS. Table 1 summarizes the most widespread ones.

**Table 1.** Summary of the main process and product certifications.

| Standards | Purposes |
|---|---|
| *Process Standards* | |
| ISO 9001:2015 | International standard that defines the guidelines for firms that want to organize their business processes implementing a Quality Management System. |
| ISO 14001:2015 | International standard that defines the guidelines for integrating environmental principles in their business processes with the aim to implement an Environmental Management System. |
| EMAS III | European standard that defines the guidelines for integrating environmental principles in their business processes with the aim to implement an Environmental Management System. |
| OHSAS 18001:2007 | British standard that permits to manage organizational health and safety risks in firms, developing a Health and Safety Management System |
| ISO 45001:2018 | International standard that permits to manage organizational ethical, health and safety risks in firms, developing an Ethical and Health and Safety Management System |
| SA 8000:2014 | International multi-stakeholder standard that permits to develop, maintain, and apply socially acceptable practices in the workplace, with the aim to implement an Ethical Management System |
| *Product Standards* | |
| Environmental Product Declarations (EDP) | Following the lines of the international ISO 14026:2006 standard, it defines Type III environmental declarations. It is verified by a third-party certification body and has a comparative purpose |
| ISO 14040:2006 | It describes the principles and framework for developing a life cycle assessment (LCA) of products, evaluating their environmental impact from cradle to grave. |
| Forest Stewardship Council certification (FSC) | FSC is an independent, non-governmental, not-for-profit organization established to promote the responsible management of the world's forests. This standard guarantees that the FSC-labeled product has come from a forest and supply chain that is managed responsibly. It defines Type III environmental declarations. It is verified by a third-party certification body and has a comparative purpose |
| Carbon Footprint (CF) | CF is a life cycle assessment based on the Global Warming Indicator which communicates the total amount of greenhouse gas emissions linked to a product throughout its supply chain. |
| UNI 11674:2017 | Italian standard that defines the requirements for the determination of the Italian origin of furniture from a sustainability perspective. It claims that the significant phases must be carried out on Italian territory and the finished products must guarantee certain minimum safety and durability requirements. |

The International Organization for Standardization (ISO) 9001 standard provides specific guidelines for companies who want to adopt a quality system to provide quality assurance to their business counterparts or final customers and to achieve zero-defect products [73]. Environmental Management Systems, designed and certified according to the International ISO 14001 standard and the European Eco-Management and Audit Scheme (EMAS), are usually used for integrating environmental protection policies and programs into the company, and for reaching the International and European goals of sustainable development [74,75]. Regarding the certification of business safety requirements, the OHSAS 18001 standard is aimed to control Organizational Health and Safety (OHS) risks in a proactive way. More recently, the ISO committee has published a new standard (ISO 45001) applicable to any organization moving towards the establishment of an internationally recognized health and safety management system [76]. Finally, in order to translate the Corporate Social Responsibility (CSR) agenda into organizational settings, the main international standard is the Social Accountability 8000 (SA 8000), which is considered a multi-stakeholder standard that helps organizations to develop, maintain, and apply socially acceptable practices in the workplace [77].

As for Product Certifications, Environmental Product Declarations (EDPs) aim to promote the sales of pro-environmental products. However, several different labels with significantly diverging

requirements exist. The requirements may be based on a life cycle assessment of the product following the lines of the ISO 14040 standard or may focus on discrete issues such as the quality of raw material and recyclability [78].

Focusing on the wood furniture sector, there is the European Ecolabel declaration that informs consumers about the intrinsic and extrinsic properties of a product. It is based on the compliance of some product and process requisites. Ecolabel copes with information asymmetries in the commodity chains between producers and consumers regarding the environmentally friendly nature of the production process [79]. This label is voluntary in nature, since its adoption is free and depends on the willingness of the producer. Another voluntary certification, which provides firms with standards and compliance mechanisms for sustainability and socially responsible practices is the Forest Stewardship Council certification (FSC) for sustainable wood, defined as the "gold" standard for wood sourced from well-managed forests [80]. The FSC label could be used by luxury furniture companies to include environmental aspects in the basic stages of the life cycle products (from design to commercialization, up to the final disposal). In this respect, there is also the Carbon Footprint (CF) ecolabel, used for assessing the global warming indicator of a given product. This label contributes to enhance the companies' ability to improve their products and process efficiency from an environmental standpoint and, thus, to accelerate their entrance into the market for green products [29]. Strictly linked to the concept of eco-design, which has been increasingly considered by furniture producers as a great opportunity for differentiating their products [29,81], a further standard can be implemented, namely EcoDesign or Design for the Environment (DfE), which integrates multiple aspects of both design and environmental considerations [27]. Finally, for the wood furniture sector, there is the Italian "Made in Italy" labeling that certifies, according to the UNI 11674:2017 standard, the requirements for the determination of the Italian origin of furniture from a sustainability perspective. Italy has a traditional competitive advantage in the production of luxury products and the Italian luxury firms with a rich cultural heritage embedded within their brands capture loyal customers and remarkable profits from all over the world. Their luxury brands are deeply rooted in their design, quality, artisanship, and service and it has often taken decades to build their reputation [82]. Notably, the UNI 11674 standard helps certifying the features of Italian luxury products claiming that the significant phases of production (i.e., manufacturing of semi-finished and finished products, final assembly and packaging) must be carried out in Italy and the finished products must guarantee certain minimum safety, and durability requirements according to relevant technical standards (www.uni.com).

## 3. Research Method

The research method for this study is based on a qualitative multiple case study. According to Creswell [83] (p. 97), this method "explores a real-life, contemporary bounded system (a case) or multiple bounded systems (cases) over time, through detailed, in-depth data collection involving multiple sources of information [ ... ] and reports a case description and case themes".

This approach is very useful to understand contemporary phenomena and practices [84,85] and to provide background material to actual issues which are still unknown [86]. Case studies, indeed, have been used by researchers not only to test theory [87], but also to describe specific contexts [88] as well as to develop theory [86,89].

In this study, it was adopted mainly for render descriptions about the extent to which Circular Economy and sustainability practices are implemented by companies operating within the luxury furniture sector. In particular, the multiple case study [84] was adopted because it allows a clearer understanding and characterization of the investigated phenomenon [90], by comparing similarities and differences emerging from the analysis [91].

The case study method is also known as a "triangulated" research strategy as it uses information collected in different times and contexts to corroborate the overall research findings. The triangulation of data provides greater depth to the study of phenomena from different perspectives [92,93]. Therefore, the use of different detection tools is recommended as a requirement for information

reliability and the overall quality of the research [94]. Accordingly, in this study, even when direct interviews represented the primary source of data collection, a review of the companies' websites and their profiles on different social networks has been carried out together with a careful analysis of further information provided by managers interviewed or sourced by the authors themselves. This involved a widespread search of documents, such as industry conference proceedings, papers, and consultant's reports on each case.

*Data Collection and Analysis*

Since the definition of luxury is still blurry, luxury furniture firms for this study were originally selected according to the following criteria:

- industry: manufacturing companies operating in the furniture sector, more specifically in the subsector of furniture and furnishing accessories, such as tables, chairs, shelves, display cases, magazine racks, umbrella holders and other accessories;
- market: companies operating on a global scale;
- firm size: small- and medium-sized companies with a turnover not exceeding €50 million and a number of employees less than 250 [95];
- luxury orientation: companies present themselves and are perceived as examples of excellence in their sector, by producing luxury items which provide unique features and high-quality finishing.

Four companies accepted to take part in the research, despite a greater number of firms having been originally contacted. This is in agreement with Yin's guidelines [84], suggesting that the number of units to be analyzed in a multiple case study is between four and twelve. Notably, with less than four cases, it is difficult to build a structured theory and to come to a "replication logic" [84].

Primary data were collected using a semi-structured questionnaire that was divided into three sections, as follows: the first part covered the background of the company (i.e., size, number of workers, types of products, etc.). The second part assessed the company's approach towards sustainability and Circular Economy, particularly on implemented practices and related enabling/hindering factors. Finally, the third part investigated the use of certifications and their role within both the communication strategies and the overall sustainable approach of the companies analyzed.

Prior to each interview, publicly available secondary material and promotional information provided by each board were reviewed to increase the researchers' familiarity with the case.

The questionnaire was directly submitted to the companies. Each interview lasted for, on average, two hours.

The cases were analyzed following the Eisenhardt's [89] guidelines of within-case and cross-case analysis. Each case was deeply investigated to gain a rich understanding of the main practices developed to move towards Circular Economy. The cases were then compared to analyze similarities and differences and to gain greater understanding of the topic under investigation.

During the study, different methods for improving the quality of the research were adopted. First, some experts were involved to support the selection of case studies. Second, interviews were conducted by the same researcher to reduce the role of bias [96] and respondents were given the opportunity to provide feedback on initial findings to reinforce the overall reliability of information. Moreover, to evaluate the validity and truthfulness of most of the propositions identified for the questionnaire, some answers were evaluated using the 5-point Likert scale, a technique widely used for measuring opinions and attitudes [97].

## 4. Results

*4.1. Company Profiles*

The companies analyzed are manufacturers, producing luxury furniture and furnishing accessories originating from all over Italy, particularly North and Central Italy, where some of the most

important Italian furniture districts are located (such as Lombardy, Marche, Piedmont, Apulia and the Triveneto-Area regions). All companies operate on international markets. As for the dimension, according to the European Recommendation 2003/361 [95], only one company can be defined as medium-sized, with a number of employees between 50 and 250 and an annual turnover of over €10 million and less than €50 million. The other companies cannot be univocally defined in terms of size, having cross-categorical parameters concerning the number of employees or annual income. A general profile of the companies is shown in Table 2.

**Table 2.** Socio-demographic characteristics of the companies.

| | C1 | C2 | C3 | C4 |
|---|---|---|---|---|
| Position held | Marketing and Communication Manager | Marketing and Communication Manager | Sales Manager | Chief Executive Officer |
| Headquarter | Northern Italy | Northern Italy | Central Italy | Central Italy |
| Reference Markets | Italy, Europe, and Asia | Italy, Europe, USA, and Asia | Italy, Europe, USA, and Asia (Shanghai, Dubai, China) | Italy, Europe, USA (Canada), and Asia (China, Japan) |
| Product Typology | Multiproduct | Multiproduct | Living | Multiproduct |
| Employees | 90 | 78 | 25 | 50 |
| Turnover: | €60 million | €16.5 million | €15 million | €8 million |
| Turnover (Italy) (%) | 30.0 | 25.0 | 80.0 | 35.0 |
| Turnover (European Union-EU) (%) * | 40.0 | 50.0 | 10.0 | 35.0 |
| Turnover (Extra-UE) (%) | 30.0 | 25.0 | 10.0 | 30.0 |
| Dimension (2003/361/CE) | Medium–Large Company | Medium Company | Small–Medium Company | Small–Medium Company |

* The turnover percentage for Europe does not include Italy.

*4.2. The Companies' Approach Towards Circular Economy*

The companies interviewed are all aware of Circular Economy and related issues, above all C4, a multiproduct furniture company, located in the Marche region and specializing in the production of luxury glass furniture and furnishing accessories (e.g., tables, chairs, mirrors, showcases, umbrella stands, magazine racks, and shelves). Their CEO declared to be aware of the topic as well as highly informed of its related activities. Despite the high interest emerging in these issues, only one company declared to be actually involved in Circular Economy projects promoted by national or international public institutions.

Practices related to the 3R principles are only partially implemented (Table 3). Reduction activities involved three companies (C1, C3, C4), especially concerning the reduction of raw materials per product unit and the overall reduction of raw materials and energy. Initiatives for enhancing energy efficiency and the use of renewable energy have been highly implemented by company C1.

As for reuse activities, these are scarcely carried out by the companies. C1 declared to make moderate reuse of equipment cleaning materials, while C3 is moderately involved in reusing product packaging materials.

The recycling activities of waste produced in the manufacturing process are employed very little by two companies, moderately in one company and highly in the last. Only company C3 declared to significantly recycle waste products coming from consumers, while company C4 intensively reprocesses waste and garbage to manufacture new products.

Except for C2, which seems to be the least involved company in Circular Economy practices, it emerges that there is more attention and willingness towards its adoption in the future, even if, companies sometimes

declared decreasing plans for future implementation practices from their current ones. For example, C1 plans a reduction in the use of renewable energy and the re-use of equipment cleaning materials, while C4 plans a reduction in the re-use of product packaging materials and equipment cleaning materials as well as in the recycle of waste produced in the manufacturing process. This seems to be contradictory with respect to the overall interest of the companies towards circularity. However, it should be noted that the companies themselves say that they intend to increase their commitment in many other circular practices, thus suggesting that the reduction plans could reflect a general search for balance between the costs and benefits associated with the general development of a circular business model.

**Table 3.** Current and future implementation of Circular Economy practices *.

|  |  | Current Implementation | | | | Future Implementation | | | |
|---|---|---|---|---|---|---|---|---|---|
|  |  | **C1** | **C2** | **C3** | **C4** | **C1** | **C2** | **C3** | **C4** |
| *Reduce* | Reduction of raw materials per product unit | 3 | 1 | 5 | 3 | 4 | 2 | 3 | 5 |
|  | Overall reduction of raw materials and energy | 3 | 1 | 3 | 5 | 4 | 2 | 4 | 5 |
|  | Use of renewable energy | 4 | 1 | 1 | 3 | 3 | 2 | 4 | 5 |
|  | Initiatives for enhancing energy efficiency of production equipment | 4 | 2 | 1 | 3 | 4 | 2 | 1 | 3 |
| *Reuse* | Re-use of product packaging materials | 2 | 2 | 3 | 2 | 3 | 2 | 3 | 1 |
|  | Re-use of equipment cleaning materials | 3 | 1 | 1 | 2 | 2 | 2 | 3 | 1 |
|  | Re-use of leftover material to manufacture other products | 2 | 2 | 1 | 2 | 2 | 2 | 3 | 5 |
| *Recycle* | Recycle of waste produced in the manufacturing process | 2 | 2 | 3 | 4 | 4 | 2 | 3 | 3 |
|  | Recycle of waste products from consumers (e.g., out of date and returned products, … ) | 1 | 1 | 5 | 1 | 2 | 2 | 3 | 1 |
|  | Reprocessing of waste and garbage | 2 | 1 | 1 | 1 | 2 | 2 | 1 | 1 |
|  | Reprocessing of waste and garbage to manufacture new products | 1 | 1 | 1 | 5 | 2 | 1 | 1 | 5 |

\* The list of Circular Economy practices was based on Zeng et al. [69]. Answers were evaluated on a 5-point Likert scale (1 = not at all important, 2 = very little important, 3 = moderately important, 4 = high important, 5 = very high important).

Motivations related to the reduction of the environmental impact of manufacturing processes are particularly relevant for companies to adopt Circular Economy practices. Furthermore, economic motivations emerged as critical, such as the increasing efficiency linked to the reduction of total costs and the possibility to achieve growing sales, especially among consumers involved in sustainability and related issues. Three companies believed that such practices are very important to obtain a long-term competitive advantage, while low interest has been devoted to the social motivations related to the health conditions of the population and the reduction of the unemployment rate. Only companies C1 and C3 respectively considered these issues as critical motivators for implementing Circular Economy practices (Table 4).

**Table 4.** Motivations towards CE practices *.

|  | **C1** | **C2** | **C3** | **C4** |
|---|---|---|---|---|
| Reduction of the environmental impact of manufacturing processes | 4 | 1 | 4 | 5 |
| Reduction of risks related to dependence on raw materials | 3 | 1 | 5 | 3 |
| Gaining a competitive advantage compared to competitors | 4 | 1 | 5 | 5 |
| Greater possibilities in order to obtain public funding | 3 | 1 | 1 | 4 |
| Reduction of the total amount of costs, thus enhancing efficiency | 4 | 1 | 3 | 4 |
| Improve people and workers' health conditions | 4 | 1 | 2 | 3 |
| Increasing total amount of sales, especially among consumers aware of sustainability and related issues | 3 | 1 | 5 | 5 |
| New professional figures to be created and reduction of the unemployment rate | 3 | 1 | 5 | 3 |

\* Answers were evaluated on a 5-point Likert scale (1 = not at all important, 2 = very little important, 3 = indifferent, 4 = enough important, 5 = very important).

Regarding the enabling factors, two companies recognized fiscal and economic incentives as crucial factors supporting the implementation of Circular Economy practices. An adequate level of consumer awareness about environmental issues is also considered as very relevant. Despite the emergence of different opinions, an efficient differentiated waste collection system and the use of artificial intelligent systems in production/distribution processes have been considered as important by two of four companies analyzed (Table 5).

**Table 5.** Enabling factors for implementing CE practices *.

|  | C1 | C2 | C3 | C4 |
|---|---|---|---|---|
| Fiscal and economic incentives for investments in R&D | 5 | 3 | 1 | 5 |
| Efficient differentiated waste collection system | 5 | 3 | 5 | 3 |
| Possible use of artificial intelligence systems in production/distribution processes | 5 | 1 | 5 | 3 |
| Adequate degree of awareness about environmental issues among consumers | 5 | 2 | 5 | 5 |

* Answers were evaluated on a 5-point Likert scale (1 = not at all important, 2 = very little important, 3 = indifferent, 4 = enough important, 5 = very important).

Different factors have also been considered to explain what hinders the adoption of circular practices. Companies underlined the difficulties in reusing some special materials used in manufacturing processes as well as the difficulties related to the reconversion of final products into new ones. While the existence of inefficient waste collection systems has not been considered as a significant obstacle by the companies, this is not valid for expensive waste disposal processes, thus revealing the importance of the economic limitations. The lack of financial resources to invest in R&D activities emerged as a critical barrier only for one company analyzed (Table 6).

**Table 6.** Factors hindering the implementation of CE practices *.

|  | C1 | C2 | C3 | C4 |
|---|---|---|---|---|
| Difficulties related to the type of material used in manufacturing processes | 5 | 5 | 3 | 5 |
| Difficulties related to final products | 5 | 5 | 3 | 5 |
| Lack of financial resources for R&D investments | 3 | 1 | 1 | 5 |
| Inexpensive waste disposal processes | 4 | 3 | 5 | 5 |
| Inefficient waste collection system | 3 | 2 | 3 | 3 |

* Answers were evaluated on a 5-point Likert scale (1 = not at all important, 2 = very little important, 3 = indifferent, 4 = enough important, 5 = very important).

*4.3. The Adoption of Product and Process Certifications*

Despite the interest in adopting product and process certifications for the near future, such standards are very little employed by companies interviewed (Table 7). In detail, two companies (C2 and C4) implemented the international standard UNI EN ISO 9001 concerning the adoption of a quality system aimed at achieving a zero-defect product objective and providing quality management practices to customers and business counterparts. Company C2 also adopts the OHSAS 18001 standard for business safety requirements.

Product certifications, on the other hand, are not used by the companies, except for the FSC voluntary certification for sustainable wood that is adopted only by company C1 (that is the larger company analyzed). Surprisingly, no company declared to adopt the Italian "Made in Italy" labeling that certifies the Italian origin of furniture from a sustainability perspective. This is rather unexpected because all four companies produce high quality products, which are internationally recognized for their high design content as well as for the exclusivity of materials and related production processes.

**Table 7.** Implementation of product and process certifications.

|  | C1 | C2 | C3 | C4 |
|---|---|---|---|---|
| Product certification | Yes | No, but we're interested in pursuing it in the near future | No, but we're interested in pursuing it in the near future | No, but we're interested in pursuing it in the near future |
| Type of Product certification | FSC | | | |
| Process certification | No, but we're interested in pursuing it in the near future | Yes | No, but we're interested in pursuing it in the near future | Yes |
| Type of Process certification | | UNI EN ISO 9001-OHSAS 18001 | | UNI EN ISO 9001 |

Companies were asked whether they were interested in product and process certification adoption in the near future (Table 8). They seem to be highly aware of the social and environmental benefits associated with their adoption. The possibilities to assure compliance with environmental standards, to improve the workers' safety and to develop a socially sustainable strategy are recognized among the most important factors motivating the adoption of product and process certifications for the future. Meanwhile, economic benefits linked to the overall improvement of process efficiency emerges as a potential stimulus. The communicative advantages associated with such certifications appears interesting too. Companies C1 and C3, in particular, gave a lot of importance to the possibility of increasing customer loyalty, improving corporate image and entering new market segments thanks to the use of international standards. This reveals the existence of a high degree of knowledge and awareness about product/process certifications which, actually, still does not reflect an adequate implementation of related standards within the companies analyzed.

**Table 8.** Motivations towards product/process certification adoption *.

|  | C1 | C2 | C3 | C4 |
|---|---|---|---|---|
| Total costs reduction | 3 | 1 | 5 | 3 |
| Achievement of high-quality standards of products compared with competitors | 4 | 1 | 5 | 3 |
| More effective monitoring of manufacturing processes with efficiency improvement | 5 | 2 | 5 | 5 |
| Increasing customer loyalty | 4 | 2 | 5 | 3 |
| Possibility of entering new markets | 5 | 2 | 5 | 3 |
| Improving corporate image | 5 | 3 | 5 | 2 |
| Improving workers' safety at workplaces | 5 | 3 | 5 | 3 |
| Compliance with environmental legal standards | 5 | 3 | 5 | 3 |
| Developing a socially sustainable strategy | 5 | 3 | 5 | 3 |
| Possibility of using funding from national/supranational public authorities | 5 | 2 | 1 | 3 |

* Answers were evaluated on a 5-point Likert scale (1 = not at all important, 2 = very little important, 3 = indifferent, 4 = enough important, 5 = very important).

Finally, we considered the communication activities which companies declared to use or to be interested in using for communicating their attention towards Circular Economy and their general commitment towards sustainability. As shown in Table 9, companies' websites, printed catalogues, brochures and participation to fairs are considered enough or highly important activities. Similarly, the role of product and packaging labels as communication tools has been stressed by two companies.

Notably, the interest towards collaborations with universities and/or research centers not only reveals their potential as communication tools, but also confirms the companies' willingness to be embedded in a circular model, by supporting the growth of adequate skills and competencies.

**Table 9.** Tools for communicating CE and sustainability practices *.

|  | C1 | C2 | C3 | C4 |
|---|---|---|---|---|
| Drafting of Sustainability Budget | 5 | 1 | 1 | 2 |
| Product label information | 5 | 1 | 5 | 2 |
| Product packaging information | 5 | 1 | 5 | 2 |
| Participation at sectoral fairs | 4 | 1 | 5 | 2 |
| Participation at workshops and events with reference to sustainability and related issues | 2 | 1 | 5 | 4 |
| Use of website (also for publicizing information related to product and process certifications, environmental and energetic management systems) | 4 | 1 | 5 | 5 |
| Use of company's website | 4 | 1 | 5 | 5 |
| Participation in forums, blogs, communities, online discussion groups related to sustainability issues | 2 | 1 | 5 | 5 |
| Collaborations with universities, research centers in order to realize/develop projects related to circular economy and sustainability issues | 4 | 1 | 5 | 5 |
| Use of companies' catalogues/product brochures and instruction manuals | 5 | 1 | 5 | 5 |
| Periodic publications on dedicated magazines | 2 | 1 | 4 | 1 |
| Mass-media advertising (TV, radio, press) | 1 | 1 | 2 | 1 |
| Periodic reports and bulletins that analyze/certify environmental ratings, compliance with certifications, … | 2 | 1 | 5 | 1 |

* Answers were evaluated on a 5-point Likert scale (1 = not at all important, 2 = very little important, 3 = indifferent, 4 = enough important, 5 = very important).

## 5. Discussions and Implications

The present study provides evidence about the adoption of Circular Economy and sustainable practices in the luxury furniture sector.

A worthy degree of awareness and knowledge of Circular Economy principles seems to emerge from the analysis. Two companies declared to know well of the concept of Circular Economy and would like to deepen its principles, while one company declared to be well informed about the concept, as it has already gathered much information and developed adequate internal training aimed at promoting a circular model.

This is in line with previous literature analysis, which highlights the increasing attention towards Circular Economy among all companies [5,12,13], including the luxury ones [21].

Despite this attention, the analyzed companies declared to still be little involved in circular practices, especially concerning reuse and recycle actions, which are particularly important under a circular perspective. Consistent with previous findings of Yuan and colleagues [33], this exploratory study seems to reveal a greater attention of companies towards reducing activities, concerning the use of raw materials or energy, which are directly related to company efficiency and economic benefits. Motivations that push the luxury companies to adopt Circular Economy practices, in fact, would be mainly of an economic and environmental nature. This reveals a partial understanding of the potential advantages linked to Circular Economy. In actuality, managers tend to be focused on perceived economic advantages, such as process efficiency as well as costs reduction, and environmental advantages linked to such practices, but they still underestimate the potential social impact, which can derive from their adoption in terms of the reduction of the unemployment rates or the general improvement of social well-being. Hence, several initiatives are needed to reduce this gap and to improve the companies' awareness of both Circular Economy practices and related consequences.

Similarly, regarding the role of product and process certifications, despite their slight adoption, the research shows a good level of awareness by luxury companies about their opportunities. Managers interviewed recognized the economic and social benefits associated with their implementation as well as marketing tools to improve companies' ability to enter new markets and to increase consumer loyalty and brand awareness. This suggests the overall appreciation by luxury companies of both efficacy and efficiency advantages linked to certifications. However, some initiatives—detailed later in the paper—could be useful to help managers in the implementation phase of such standards. Therefore, further implications can be proposed for both private companies and public institutions.

Because circularity requires regenerative and restorative practices [6], companies should develop strategies that allow reducing as well reusing and recycling of components and materials, from the design phase. Referring to the specific context of the luxury furniture sector, designers play a crucial role as their creative freedom affects materials and characteristics of final products. In this regard, it would be desirable for these processes to be based on the use of recyclable materials, which encourages the reuse of products also in the end-of-life phase. Internal training as well as informal and planned meeting for discussing the circularity issues could be useful for sharing the diffusion of a new business model that requires an overall cultural change and a general involvement of all skills within the company, including designers. Moreover, internal communication could become critical to inform the personnel about the opportunities of circularity and its economic, environmental and social advantages. Periodic reports, along with the use of indicators, can be helpful for summarizing Circular Economy benefits and operationalizing its results.

Circular Economy also enhances the introduction of a new consumption model where property is replaced by access [28]. This model should guide the transformation of consumer-owners into consumer-users, being aware that after the use of the product, they could return it, to be reused or recycled. Within the luxury market, this approach could be in contrast to a typical upper-class consumer behavior aimed at extending product life as much as possible [21]. In fact, its high economic and affective value, could sometimes result in the consumer's willingness to "pass" the luxury good to the next generation. Regardless, the transformation from consumer-owners to consumer-users implies more information and awareness about Circular Economy benefits. Actually, an adequate level of consumer awareness of Circular Economy has been recognized as critical by the companies interviewed for enabling Circular Economy implementation. Moreover, prior research also stressed this issue, by arguing that luxury purchasers are still less concerned by sustainability preoccupations, despite their increasing attention towards sustainability [21]. As argued by Kapferer and Michaut-Denizeaut [21], indeed, upper-class customers are often "interested mostly in the design (how the furniture looks/feels and what impression creates in their living environment), followed by customization [ . . . .] then of course the materials and structural design". Therefore, it should be important to improve adequate training for the final market, which could be promoted by both schools and local institutions (e.g., events, workshops, seminars to raise individuals' awareness on Circular Economy issues). Companies should also promote this new model of consumption, for example, by offering economic incentives stimulating consumers to give back products in the end-of-life phase. Certainly, this requires a widespread distribution network and high economic and managerial resources, which sometimes are lacking in smaller companies [98]. However, it could be recommended for reinforcing reusing practices.

Another basic principle of Circular Economy concerns the reverse cycles. To create value from used materials and products, it is necessary to collect them and take them back to their origins [28]. Therefore, companies could improve their ability to implement a reverse logistic and an efficient system of waste and product leftovers treatment that allows the return of such materials to the market. Furniture companies, as well as others operating in similar contexts, could move to implement such activities, as they can, by improving their ability to recover components into new products addressing the specific market needs. This requires financial and organizational resources, since important investments could be necessary for processing reused materials and waste disposal. The role of economic limitations and the lack of financial resources as potential obstacles to Circular Economy implementation clearly emerged in this study. In this regard, the role of governments and public institutions could be critical, as they could provide suitable economic and financial incentives and measures to support the companies' efforts, in addition to a merely administrative approach [15]. Technology and investments in R&D, in fact, play a crucial role in the development of Circular Economy as they allow the implementation of new innovative and creative processes by companies, but their actual management requires financial investments which, sometimes, could discourage firms, especially those of small size that are usually facing resource scarcity issues.

Finally, a further suggestion can be proposed from the analysis of product and process certifications. Notwithstanding the recognition of potential benefits linked to the use of international standards of certification, they are very little applied by the luxury companies interviewed. Literature shows that such standards may be particularly helpful to aid managers in implementing environmentally sustainable practices, thus moving towards a wider Circular Economy model [34]. Thus, they should be improved to operationalize the general attitude towards circularity, to strengthen the sustainable approach within the company by engaging all skills and internal figures, and to communicate such an approach to consumers and other stakeholders.

Managers interviewed did not give detailed explanations about the reasons underlying the low use of certifications, saying that they are working to implement them in the near future. Certainly, this deserves further analysis. However, based on prior research [73,99], it is likely to suppose that the adoption of product and process certifications tends to be very difficult for companies mainly for high implementation and maintenance costs, but also for the high bureaucracy and organizational complexity that they brought with them. The implementation of IMSs and product certifications requires time and figures dedicated exclusively to their management and this could be the most relevant limitation to their implementation for companies [100]. In order to overcome such limits, a change in companies' culture and in top management involvement could be recommended. Companies should see beyond the initial barriers to the adoption of these certifications, trying to think of them as a medium–long-term investment that will lead to organizational and environmental improvements, as well as to brand image and competitiveness advances [68]. In this respect, useful suggestions also come from the study of González-García et al. [27], which demonstrated how the implementation of DfE in the development of furniture products can help to introduce innovations in the production process, which allows the reduction of the environmental impact as well as improvement for the firms' overall efficiency in a short period of time.

Moreover, product and process certifications could be helpful for supporting the luxury companies' ability to develop a communication strategy aimed at highlighting their commitment to sustainability, going beyond the main objective of providing luxury experiences or host events [21]. Luxury companies, indeed, should communicate more about their sustainable efforts, by informing consumers and stakeholders. Thus, putting environmental issues on product labels and packages should help luxury companies to communicate not only the product features, by using truthful claims and messages, but also their sustainable orientation and the overall characteristics of the territory in which they operate. In this study, the two companies interviewed clearly recognized the role of product labels as communication tools for sharing sustainable values.

Crucially, with respect to communications strategies, luxury companies could also improve their performances by developing ad hoc activities, which could have a relevant impact in the long run, such as, special days in which they open their workshops to the public to show how products are made and how materials are used and recycled within the company.

Overall, a further suggestion could be derived from the literature, which highlights the importance of a favorable system condition. In particular, Schuler and Buehlmann [101] demonstrated the importance of centers of excellence or industry clusters, based on strategic partnerships between manufacturers and suppliers, customers, supporting institutions, and other stakeholders, in order to enhance the development of a value-added products culture surrounding the furniture sector. Industry clusters could be critical for establishing and maintaining a global competitive position [102]. In Northern Italy, for example, a tight cluster of chair manufacturing-related businesses produce 30% of the world's annual wooden chair production [101]. As highlighted by Schuler and Buehlmann [101], each actor of the cluster reinforces the other, thus strengthening the firms' competitiveness, thanks to the overall sharing of resources, competencies and skills.

The collaboration between value chains and sectors could be very important for establishing a large-scale circular system [6,28]. Relationships with other firms could provide some facilities, and help companies for product development and information sharing, as well as sectoral standards adoption.

Particularly, small businesses may need guidance in areas such as the recovery, reuse and remanufacture of goods and materials [37]. Moreover, within a network perspective, the waste produced by a company could become the raw material for another and companies in the same cluster could share materials and energy flows, thus enhancing both cooperation and resource exploitation reduction.

## 6. Conclusions, Limitations and Future Research Directions

This explorative study adds further evidence to the existing literature about Circular Economy principles and characteristics in the luxury furniture sector. The results show that Circular Economy applied to this sector is yet a new concept towards which companies would be increasingly addressing their attention. The case studies analyzed, in fact, seem very interested in Circular Economy, even if they would need to develop further knowledge about it. Notwithstanding their interest, a potential gap seems to emerge between the positive attitude towards Circular Economy and their practical implementation since Circular Economy practices adoption still seems limited and mainly related only to reducing actions. Similar findings emerged with respect to product and process certifications. The companies interviewed are aware of their potential advantages, both economically and communicatively, but they admitted to using them very slightly. Therefore, some gaps that would be worthy to be explored in future research emerged. Further qualitative analysis could be drawn to proceed with explorative purposes, as Circular Economy is still incipient and the transition to a more structured Circular Economy model seems to be in its initial stage [28]. Further research could be conducted, particularly involving Circular Economy applications on the end-of-life phase of the product, in which very scarce knowledge and practices emerged from this study. Moreover, an in-depth analysis of both organizational and structural conditions that are needed within specific furniture districts should be interesting to promote the development of Circular Economy practices.

Other research perspectives come from the limitations of our study, mainly due to the multiple case study adoption. The results of this research are not characterized by generalizability [103]. Therefore, a greater number of case studies, along with the development of a quantitative survey, could be recommended for the future, in order to provide a better understanding of the concept of Circular Economy and practices within the furniture sector. A comparison with other industries could also be useful for revealing peculiarities and differences among various market contexts. Finally, as previously noted [104,105], future research could be deepened by searching for criteria and indicators for assessing the level of circularity of both products and companies.

**Author Contributions:** Data curation for the research paper has been done by M.B. and L.B., as well as the methodology section, in collaboration with E.S. The writing of the original draft has been done by E.S. and M.B., while subsequently E.S. and L.B. did the final review and editing of the paper.

**Funding:** This research received no external funding.

**Conflicts of Interest:** The authors declare no conflict of interest.

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
