# Peer review of "Sustainability and Quality Management in the Italian Luxury Furniture Sector: A Circular Economy Perspective"

_sustainability, doi:10.3390/su11113089_

Round 1
Reviewer 1 Report
This an interesting theme; the circular economy is part of the solution for the sustainability of our economic development model. The authors analyse the knowledge, attitude, motivation and implementation of the Circular Economy concepts (crossing them with quality management and certification issues) in the Italian luxury furniture business.
This exploratory paper has the potential to provide useful insights regarding the implementation of the Circular Economy concepts in a relevant economic activity. At this point, however, the paper is yet far from realising that potential. I have concerns (and suggestions) regarding both the paper’s substance and the form.
Please see my full revision text in the attached file.

Author Response
Notes on the revision of the manuscript ID: sustainability-505455
Title
SUSTAINABILITY AND QUALITY MANAGEMENT IN THE LUXURY FURNITURE BUSINESS: A CIRCULAR ECONOMY PERSPECTIVE
Dear Editor and Reviewers,
We wish to thank you for the opportunity to revise and resubmit our paper to Sustainability. We are glad you agree that the topic is worthy of investigation. We read your comments carefully and with full consideration in order to guide our efforts to improve the paper.
Below, we will indicate point-by-point how we addressed the Reviewers’ comments during our revision.
Thank you for providing detailed and challenging reviews that have helped immensely in our efforts to realize the potential of this research.
------------
Responses to comments of Reviewer 1
General comment:
This an interesting theme; the circular economy is part of the solution for the sustainability of our economic development model. The authors analyze the knowledge, attitude, motivation and implementation of the Circular Economy concepts (crossing them with quality management and certification issues) in the Italian luxury furniture business.
This exploratory paper has the potential to provide useful insights regarding the implementation of the Circular Economy concepts in a relevant economic activity. At this point, however, the paper is yet far from realizing that potential. I have concerns (and suggestions) regarding both the paper’s substance and the form.
Response to the general comment:
Dear Reviewer,
Thank you very much for your review. We are glad that you found our paper as potentially interesting for the scientific debate.
We feel that incorporating your comments into our paper has significantly benefited our work. Below, please find a point-by-point reply on how we dealt with each of your recommendations during the revision process.
AREA 1: In what regards the content
Comment 1: Title: It is relevant to add the Italian context of the study on the title. Instead of the current title, please consider “Sustainability and Quality Management in the Italian Luxury Furniture Business: A Circular Economy Perspective”.
Response to comment 1:
Thank you very much for this observation. As suggested, the title has been modified as follows: “Sustainability and Quality Management in the Italian Luxury Furniture Business: A Circular Economy Perspective„
Comment 2: Keywords: I would add luxury furniture and Italy to the keywords, as they would be relevant in the paper’s indexation.
Response to comment 2:
Thanks for this suggestion. “Luxury furniture„ and “Italy„ have been added to the list of keywords.
Overall critique:
Comment 3a: I suggest this paper is repositioned as a first step, preparing a second stage with a wider number of companies reached. As a consequence, I would be less affirmative in the conclusions.
Response to Comment 3a:
Thank you for pointing out this observation. Certainly, the number of companies analysed in this study is small. While considering the urgence of a second research step based on a wider number of companies to be analyzed, we carefully revised both the Discussion and the Conclusion sections of the paper trying to be less affirmative as possible, by arguing results and implications always in a conditional forms.
The paper uses frequently loose, general, subjective concepts, e.g.:
Comment 3b(i): Page 1, line 37 “a sustainable development and a fair society”. What is a fair society? What is the criterium (or set of criteria) used, e.g. equal income, equal opportunities, an income directly related to merit?
Response to Comment 3b(i):
Thank you for this comment. We agree on the fact that a clear definition of “fair society„ is not provided in our previous manuscript. However, as this concept is rather subjective and it has not been univocally defined in the literature, we provided more explanations by citing some policy objectives pointed out by Marmot (2013). On p. 1 (lines 38-41) of the current paper, we wrote: „Thus, Circular Economy represents a new business model, useful to achieve a sustainable development [7-13], as well as a fair society, based on early childhood development interventions, increasing equality in the workplace, preventive health care and health behavior campaigns, and a better distribution of monetary income [14].“
Comment 3b(ii): Page 2, lies 58 – 62 “These companies, indeed, often use rare raw materials e.g., animal skins, gold, or gemstones); promote animal treatments (e.g., the exploitation of crocodile farms, the killing baby seals for fur); adopt manufacturing methods (e.g., mercury for tanning skins) polluting the local environment, or destruct the environment itself (e.g., use of endangered tree species by luxury furniture business, exploitation of rare water resources by luxury hotels located in poor countries).”
Response to Comment 3b(ii):
Thanks for this comment. We regret not having included any references to support this statements. After an in-depth control, it has been now included in the paper (i.e.: Kapferer and Michaut-Denizeau, 2014).
Comment 3b(iii): Page 2, lines 79 – 80: “especially when luxury brands are able to convey true values, such as quality, rarity and tradition, which are in line with sustainable objectives”. What are true values? How are they in line with sustainable objectives?
Response to comment 3b(iii):
Thank you for this comment. Taking account of the overall reviewing suggestions, we finally decided to delete this sentence, just stressing only the increasing attention towards sustainable luxury goods on the part of the customers (on p. 2, lines 59-62).
Comment 3b(iv): Page 4, line 176: “metals can be endlessly recycled”. This is an oversimplification, at best - it depends on the metal, the alloy and the specific product from which it is recovered from. There is a cost issue that has to be addresses; in some cases, the cost incurred is higher than the value recovered.
Response to comment 3b(iv):
Thank you for this important observation. Certainly, the metal recycling is a complex issue. We didn’t pay high attention to it, since it was beyond the scope of our research. However, a carefull analysis is needed before a company decides to implement metal recycling practices. Thus, we considered your suggestion by adding a brief discussion of the issue on p. 5 (lines 230-232).
Comment 3b(v): Page 5, lines 227 - 235: “Quality management tools such as Integrated Management Systems (IMS), which include the development of Quality, Environmental and Corporate Social Responsibility Management Systems (QMS – EMS – CSRMS) and Product Certifications (PC) are particularly useful… IMS combine all components of a business into one coherent system…” What about the accounting, HR, marketing and production management (ERP) systems?
Response to comment 3b(v):
Thank you very much for this comment. We agree that also the Enterprise Resource Planning systems for managing operations in accounting, human resources, marketing and production processes are important in the everyday operational management of a company, but the discussion of ERP systems was beyond the scope of our research, since ERP Systems are the practical tools for managing Quality Management Systems oriented at developing and certifying that the organizational structure of a company fulfilss environmental, ethical, quality and health and safety requirements. Therefore focusing also on these instruments would be such a wide and interesting topic that it would require a separate research paper. This could be the right suggestion for future research. However, if you consider it as necessary, we are always ready to make further changes tot he paper.
Comment 3b(vi): Page 15, line 481: “…some policies…”. Which ones? Specify.
Response to comment 3b(vi):
Thank you for pointing out this observation. As managers should be helped in the implementation of product and process certifications (and, more generally, of Circular Economy practices), we tryed to highlight at best the initiatives that could be useful in this respect. First, in the Discussion Section, we summarized the research findings and underlined the need for supporting actions aimed at enhancing the implementation of circular practices and certicifations. Later, we made an in-depth discussion of what possible actions can be done. As for certifications, in particular, initiatives are specifically argued on p. 16 (lines 646-650).
The paper is too prescriptive, without quantitative data to support the affirmations or recommendations, e.g.:
Comment 3c(i): Page 2, line 65 – 66: “Overall, the furniture sector accounts for considerable portion of global trade.” Yes? What is considerable?
Response to comment 3c(i):
Thank you for this comment. We agree that more quantitative data are necessary in order to highlight the economic and social relevance of the furniture industry, particularly in the European Union. Thus, we provided some recent data and statistical estimates, which describe the overall furniture industry and, more specifically, the luxury furniture one (p. 2, lines 83-94).
Comment 3c(ii): Page 2, lines 73 – 76: “The furniture industry appears to be particularly important from environmental standpoints since it is characterized by an intensive use of virgin raw materials and because the large use of adhesives, dyes and coating materials in furniture production results in emission of large volume of volatile organic compound in the environment annually [22].”. Does it appear to be so? Can’t we be more objective? Is this industry different of other industries? In what ways?
Response to comment 3c(ii):
Thank you for this further suggestion. On p. 3 (lines 98-102), we stressed the waste problem, by adding recent statistical data highliting the weight of the EU28 furniture waste. Moreover, we provided a discussion of both energy and raw materials used within the luxury furniture industry (p. 3, lines 103-110), as it was pointed out in Comment 4c of the present reviewing document. This is helpful to become more aware about the possible consequences arising from the use of raw materials and, above all, of resources needed in furniture production processes (which, in turn, emphasizes the potential importance of implementing circular practices within this context).
Comment 3c(iii): Page 5, lines 201 – 209. This text is particularly confusing, e.g. “…luxury firms are usually focused on high quality, so much as on family-like working conditions. Consequently, they tend to control the whole supply chain and they “are unlikely to thrive by relying on poor practices, unfair labour conditions, poor animal treatments or spoiling the environment”. Therefore, the adoption of sustainable practices seems to be less widespread among them”. Please clarify, e.g. what are family-like working conditions?
Response to comment 3c(iii):
Thank you for this comment. We acknowledged the fact that the term “family-like working conditions„ is not properly suited to express what we wanted to mean. We accepted your suggestion, by deleting the above words and writing a shorter and clearer sentence on p. 6 (lines 270-272).
Conceptual questions:
Comment 4a: The paper lacks a clear objective definition of luxury furniture. What products are included in this definition?
Response to comment 4a:
Thank you for this remark. A clear definition of luxury furniture is needed. Therefore, in the Introduction Section, we first explained the purposes for which we use the term “luxury” (p. 2, lines 59-62) and, subsequently, we provided a specification related to luxury furniture firms taken from previous literature (p. 2, lines 75-77).
Comment 4b: The paper lacks a quantitative characterization of the luxury furniture industry in the World, the European Union and in Italy. How much is produced, exported and sold? What is the volume of raw materials used and by-products, sub-products and waste generated?
Response to comment 4b:
Thank you for this comment. As for the furniture industry, more quantitative data are needed also to highlight the importance of the luxury furniture one. Despite the difficulties related to access very recent data, we used all the data available to provide a quantification of this industry (p. 2, lines 90-94). Unfortunately, after several efforts, we didn’t found detailed estimates (concerning, for example, the volume of export and products sold). We are very sorry for that. We hope that the data added will still be useful to provide a more realistic description of the luxury furniture sector.
Comment 4c:The paper lacks a conceptual model of the resources (energy and raw materials) in the luxury furniture cluster.
Response to comment 4c:
Thank your for this important observation. We acknowlege that a conceptual model of resources used in the luxury furniture cluster would have been helpful in understanding the importance of consequences related to the use of raw materials. We carefully considered your suggestion and discussed what are the main resource used in this segment with the aim of highliting the need of moving towards Circular Economy and related practices, due to their large use by companies (p. 3, lines 103-110).
Comment 4d:Page 8, lines 312 – 314: ”… firm size: small and medium-sized companies with a turnover not exceeding €50 million and a number of employees less than 250 [87], thus excluding microproduction characterizing craftsmanship;” Wrong logic: how does this excludes microproduction units?
Response to comment 4d:
Thank you for this comment. It was an obvious misprint that had to be removed before the submission. We are really sorry for that. In the current version of the paper, we deleted the wrong specification (p. 9, lines 388-389).
Comment 4e:Are the analyzed companies manufacturers or distributors of products? Do they manufacture all the products they distribute?
Response to comment 4e:
Thank you for this comment. In the current version of the manuscript, we highlighted that companies interviewed are only manufctures of products (p. 9, line 384 – p. 10, line 421).
Comment 4f:Page 9, line 357 mentions “luxury glass furniture”. Can you clarify what type of products these are? The paper had a focus on wood products up to this point.
Response to comment 4f:
Thank you for this comment. With the aim to give a clearer understanding of what products are manufactured by company C4, to which this Comment refers, we added in brackets some examples of glass products (p. 10, lines 435-436).
Comment 4g:The importance of design for sustainability (of both processes and products) needs to be stressed, given special attention in circular economy projects.
Response to comment 4g:
Thanks a lor for this further recommendation. During the revision, we improved the discussion of design and eco-design in different parts of the paper, in ordert to stress its importance within a circular economy perspective. The main changes in this direction were made in the manuscript on p. 5 and 6 (lines 242-253) and on p. 8 (lines 342-345).
Comment 4h:I would replace the expression”… their communicative role in order…” for “…marketing tool…” in line 478, page 15.
Response to comment 4h:
Thanks for this comment. The suggested change was made (p. 15, line 570), as the use of the term marketing tool appears to be more suitable to describe the potential role of product and process certification, which is not limited to communicative goals.
Comment 4i: In lines 497 - 498, the authors suggest “This model should guide the transformation of consumers-owners into consumer-users, being aware that after the use of the product they must return it, to be reused or recycled.” Do you believe that this is realistic for luxury consumers? Wouldn’t an approach of pass it to the next generation be more in line with wealthy customers? It’s sharing, not owning – the practical effect is the same, but more appealing to this kind of clients – this is my subjective opinion, I could be wrong.
Response to comment 4i:
Thank you for this comment. We acknowledge that this affirmation could be misleading thus, we accepted your suggestion, by enriching the discussion on this issue on p. 15, lines 591-595.
Comment 4j: In lines 509 – 510, the authors suggest “…offering economic incentives stimulating consumers to give back the product at its end-of-life, thus reinforcing reusing practices”. The problem is of practicability: these are SME companies yet with long life products and clients in geographically distant locations. How would a product give back at end-of-like be feasible?
Response to comment 4j:
Thanks for this comment. We agree on the fact that it is necessary a more precise clarification related to possibile initiatives aimed at stimulating consumers to be more socially responsible. Therefore, as suggested, we enriched the discussion on this topic (p., 15, lines 607-609), by highlighting the difficulties which can limit the practicability of our proposal.
Comment 4k: The connection to Italian furniture districts – line 565, page 17 – is interesting and cluster- based cooperation has great potential. This deserves further comments in this paper and research in the future.
Response to comment 4k:
Thanks for this suggestion. Indeed, the luxury furniture companies analysed are based in the furniture districts, thus we accepted your observation and stressed the importance of this clusters, mainly due to strategic partnerships that can enhance the companies‘ competitive advantage, in addition to facilitate a transition toward a circular model (pp. 16-17 lines 665-674).
Data collection and analysis:
Comment 5a: The paper summarizes the exploratory work done (based on interviews with four companies’ managers) in several tables. The variables analyzed are ordinal; you can’t calculate averages on ordinal variables. In addition, with such a reduced number of analyzed cases there is no point in summarizing results – no point in calculating measures of central tendency (mode or median, which would be acceptable in ordinal variables) or dispersion (interquartile distance, or other).
Response to comment 5a:
Thanks for this advice. In the earlier manuscript, we considered the mean values to provide some synthetic explanations of our findings. However, we certainly agree with your observations, given the qualitative nature of our study and the limited number of companies analyzed. Therefore, in the current version of the paper, we didn‘t consider mean values and they were deleted from all the Tables.
Comment 5b: Line 321, page 8: “too many cases make the data analysis complicated”. Not true.
Response to comment 5b:
Thanks for this observation. Undoubtly, the sentece you cited was ambiguous and could give rise to misunderstandings. In the revised manuscript, it has been removed and the sentence on p. 9 (lines 395-396) was changed.
Comment 5c: Page 8, lines 309 – 310: “companies operating in the furniture sector, more specifically in the subsector of furniture and furnishing accessories”. Can you provide examples of products manufactured by this sub-sector?
Response to comment 5c:
Thank you for the comment. We accepted your suggestion and provided some examples of furniture and furnishing accessories on p. 9, lines 385-386.
Comment 5d: Page 8, lines 318 – 319: “Four companies were finally used for the analysis, despite a greater number of firms were originally contacted”. Can I assume that these four were the selected as these were the only ones willing to participate in this study?
Response to comment
Thanks for this comment. We confirm that companies analysed were the only ones that expressed their willingness to partecipate in this study. This was claryfied in the paper on p. 9, line 393.
Data validation:
Comment 5e(i): In Table 3, some decrease their future implementation practices plans from their current ones. This doesn’t seem logic; is this an error or were these the answers? If these were the answers, do you have an explanation? Also, repeating a comment for Table 1, keeping the tables within the same page is a good practice.
Response to comment 5e(i):
Thanks a lot for this observation. Data shown in Table 3 are real and they need a more explanation. Thus, we included a reacher discussion in the revised paper. In detail, on p. 11, lines 452-462, we wrote: “Except for C2, which seems to be the less involved company in Circular Economy practices, it emerges more attention and willingness towards its adoption for the future, even if, sometimes, companies declared decreasing plans for future implementation practices from their current ones. For example, C1 plans a reduction in the use of renewable energies and the re-use of equipment cleaning materials, while C4 plans a reduction in the re-use of product packaging materials and equipment cleaning materials as well as in the recycle of waste produced in the manufacturing process. This seems to be contradictory with respect to the overall interest of companies towards circularity. However, it should be noted that the companies themselves say they intend to increase their commitment in many other circular practices, thus suggesting that the reduction plans could reflect a general search for balancing between costs and benefits associated with the general development of a circular business model.„
Comment 5e(i): Company 2 – C2, has all answers as 1 in Table 4 – Motivation and Table 9 – Tools for communicating. Are these answers real? Is this a problem with the data or with the company or the respondent?
Response to comment 5e(i):
Thank you for this further observation. We acknowledge that it could generate some misunderstandings, but these data were really provided by the respondent. The most acceptable explanation we can provide relates to the fact that, as stated on p. 11, line 445, C2 seems to be the less involved company in Circular Economy practices, at least for the present. Thus, values related to its current circular practices are always less than those related to future plans.
AREA 2: In the communication component of the paper
Comment 1: Style: The text is, frequently, informal and too general, e.g. in line 39, page 1, “A lot of economic and financial resources are needed to develop such a business model…” Again, in line 192, page 4, “…a lot of economic and financial resources are needed…” How much is a lot?
Response to comment 1:
Thank you for this comment. During the revision process we carefully considered this suggestion and provided several changes in order to avoid informal or too general sentences. The term “A lot„ has been removed in both the above sentences you cited.
Comment 2: Graphs: Visual aids are a key tool in communicating and summarizing information. The paper has not a single graph or image; they could be very useful explaining the concept of the Circular Economy and the flow of raw materials, products, by-product, sub-products, energy and waste.
Response to comment 2:
Thank you for this comment. We agreed on the fact that a graph/image could be very useful to explain the meaning of the concept of the Circular Economy and to summarize what are the main practices on which this approach lays its foundation. Therefore, we placed in Section 2 (p. 4) a figure that could summarize the main practices related the overall Circular Economy approach. It was retrieved from the “Soustainable Global Resources Ltd„
Comment 3: English: the English needs to be improved. I suggest a revision by a native speaker.
Response to comment 3:
Thank you for this comment. We agree on the fact that the English language of the paper needs to be improved. Hence, it was carefully revised by a native speaker to improve the overall quality and readability of the manuscript.
Other comments on form:
Comment 4a: In page 1, line 11, “… Governments’ agendas. Towards the development …” should probably be “…Governments’ agendas towards the development…”
Response to comment 4a:
Thanks for this remark. The sentence has been corrected on p. 1, line 36.
Comment 4b: In page 1, line 30, “… acid rains, depletion of resources…” should be “…acid rains and depletion of resources…”
Response to comment 4b:
Thanks for this comment. We accepted your suggestion and modified the sentence accordingly (p. 1, line 31).
Comment 4c: Page 3, line 123: “The next Section discusses…” As we are in section 1, the next section would be section 2, when the authors want to say Section 4. If you write “Section 4 discusses…” the ambiguity disappears.
Response to comment 4c:
Thank your for this comment. We wrote “Section 4„ in place of “next Section„ on p. 4, line 154.
Comment 4d: Page 3, lines 133 - 138. The full paragraph is unclear, needs rewriting.
Response to comment 4d:
Thanks for this recommendation. We agree that the sentence on p. 3 of the original manuscript was quite confusing. Consequensly, we rewrote the full sentence trying to make it more clear (p. 4, lines 164-169 of the current version).
Comment 4e: The authors use CE throughout the text, for Circular Economy. I would prefer the use of the Circular Economy, as the CE always reminds me the CE marking - a certification mark that indicates conformity with health, safety, and environmental protection standards for products sold within the European Economic Area (EEA).
Response to comment 4e:
Thank you for this observation. It is very important not to generate confusion with regards to the abbreviations, thus we accepted your suggestion and substituted “CE„ with “Circular Economy„ throughout the text.
Comment 4f: Table 1, page 6. Please avoid breaking tables into different pages.
Response to comment 4f:
Thank you for this comment. In the current paper, each Table stays in the same page, as well as captions (see also Comment 4k).
Comment 4g: Page 8, line 301: “…a review of the companies’ websites and their profiles on different social networks have been…“ should be “a review of the companies’ websites and their profiles on different social networks has been “.
Response to comment 4g:
Thank you for this remark. The sentence as been changed accordingly on p. 9, line 377.
Comment 4h: Page 9, line 346: “Italian Country” replace by “Italy”.
Response to comment 4h:
Thanks for the comment. “Italian Country” has been replaced by “Italy” on p. 10, line 422.
Comment 4i: Table 2, page 9. Please clarify if “Europe” means “European Union” or “Europe” and whether the turnover % Europe includes Italy.
Response to comment 4i:
Thank you for this remark. As the term Europe could easily generate misunderstandings, we would like to precise that within this anlysis it is referred only to the European Union. This has been claryfied in Table 2 of the current paper.
Comment 4j: Throughout the text, replace “practises” by “ practices”.
Response to comment 4j:
Thanks for this note. The word “practises„ has been replaced by “practices„ throughout the text.
Comment 4k: Table 7, pages 12 – 13: keep caption in the same page as the table.
Response to comment 4k:
Thanks for this observation. As for the tables, captions were kept in the same pages.
Comment 4l: Please check both the translation and the completeness of the Likert scales in the Tables notes - e.g. the value 3 explanation missing in Table 8; also I would prefer a 1 – not important, 2 – somewhat important 3 – important 4 – very important and 5 – extremely important (of course, the authors know best, as it’s their scale).
Response to comment 4l:
Thanks for this suggestion. We carefully checked the explanation of the Likert scale in Tables notes. We made some changes to improve the overal completeness of the explanation, while, as regarding the meaning of values, we preferred to leave 3=indifferent (in Tables 4, 5, 6, 8 and 9) as the middle value is usually considered a neutral evaluation (Likert, 1932) – and it was interpreted in that sense in our analysis.
Comment 4m: Table 7, page 13 and elsewhere in the text: replace “next future” with “near future”.
Response to comment 4m:
Thank you for this latest observation. Table 7 has been modified accordingly.
Reviewer 2 Report
On a conceptual/ methodology side, the authors reviewed many terminologies that are related to the circular economy (CE), and even propose some sort of taxonomy of CE. You’d better give your own definition of CE or tell readers which terminologies you agree(d) with the most and why exactly. Exactly the same goes for the definition of luxury: do you need for your analysis a clear economic, business, legal, and/or, say, ethical definition? Why/ why not? What's you own operational definition of luxury for the purposes of the analysis in this paper?
On an empirical note, I guess the authors are aware that many luxury clothes brands like Burberry, Chanel, Louis Vuitton burn unsold stock. Even the retail giant H&M said it now practices that. (See eg https://www.newsweek.com/luxury-brands-prefer-burn-millions-dollars-worth-clothes-over-letting-wrong-1032088)
Aside the 'reasons' those and other luxury brands might have to do so because it is outside the scope of my comments here and of the manuscript, the destruction itself as well as CO2 and other combustion gasses emmissions due to burning the unsold stock are overwhelmingly un- and even counter-sustainable. So, I think this example of luxury industry negative 'externalities' is to be taken into account or at least mentioned in the introduction. And what about the luxury furniture: ie do we have similar, comparable or contrary examples of dealing with the stock of 'elite' furniture and parts? What do we know and where do we stand there? I think these issues of the luxury business are crucial for the sustainability questions and could not be ommitted.
Author Response
Notes on the revision of the manuscript ID: sustainability-505455
Title
SUSTAINABILITY AND QUALITY MANAGEMENT IN THE LUXURY FURNITURE BUSINESS: A CIRCULAR ECONOMY PERSPECTIVE
Dear Editor and Reviewers,
We wish to thank you for the opportunity to revise and resubmit our paper to Sustainability. We are glad you agree that the topic is worthy of investigation. We read your comments carefully and with full consideration in order to guide our efforts to improve the paper.
Below, we will indicate point-by-point how we addressed the Reviewers’ comments during our revision.
Thank you for providing detailed and challenging reviews that have helped immensely in our efforts to realize the potential of this research.
------------
Response to comments of Reviewer 2
Dear Reviewer,
Thank you a lot for your review. We feel that your comments have been of great utility to improve the overall quality of our study. Below, we explain how we addressed your suggestions during the revision process.
Comment 1: On a conceptual /methodology side, the authors reviewed many terminologies that are related to the circular economy (CE), and even propose some sort of taxonomy of CE. You’d better give your own definition of CE or tell readers which terminologies you agree(d) with the most and why exactly.
Response to comment 1:
Thanks a lot for this comment. We decided to tell which definition and related terminologies we agreed the most, as in our opinion a specific definition of the concept of Circular Economy within the specific furniture sector cannot be so comprehensive of the wide variety of practices and related consequences. Notably, on p. 4, line 175, we specifically indicated our acceptance of the Ellen MacArthur Foundation’s definition, which is also the most renowned one by prior literature, given its completeness.
Comment 2: Exactly the same goes for the definition of luxury: do you need for your analysis a clear economic, business, legal, and/or, say, ethical definition? Why/why not? What’s your own operational definition of luxury for the purposes of the analysis in this paper?
Response to comment 2:
Thank you for this further suggestion, which has been similarly addressed also by Reviewer 1.
A more detailed conceptualization of the general concept of luxury and, particularly, of the luxury furniture industry has been developed in the current version of the paper. In detail, on p. 2 (lines 59-62) we provided explanation about the term “luxury”, while later in the paper (p. 2, lines 75-77) we gave a more precise definition of “luxury furniture” fitting the scope of our study.
Comment 3: On an empirical note, I guess the authors are aware that many luxury clothes brands like Burberry, Chanel, Louis Vuitton burn unsold stock. Even the retail giant H&M said it now practices that. (See eg htpps://www.newsweek.com/luxury-brands-prefer-burn-millions-dollars-worth-clothes-over-letting-wrong-1032088)
Aside the ‘reasons’ those and other luxury brands might have to do so because it is outside the scope of my comments here and of the manuscript, the destruction itself as well as CO2 and other combustion gasses emissions due to burning the unsold stock are overwhelmingly un-and even counter-sustainable. So, I think this example of luxury industry negative ‘externalities’ is to be taken into account or at least mentioned in the introduction. And what about the luxury furniture: ie do we have similar, comparable or contrary examples of dealing with the stock of ‘elite’ furniture and parts? What do we know and where do we stand there? I think these issues of the luxury business are crucial for the sustainability questions and could not be omitted.
Response to comment 3:
Thanks a lot for this comment. Certainly, it is very important to consider the bad practices of luxury companies who declared to burn their unsold stocks. Following your precious suggestion, we added a discussion on this issue, concerning both the luxury companies in general (p. 2, lines 70-72) and those operating in the furniture sector (p.2, lines 77-82), to enrich the paper’s ability to display an overall comprehension of the sustainability question within the luxury setting.
Overall, we hope that you agree the paper is stronger and we look forward to your feedback. Thank you very much for all your efforts related with our submission.
The Authors
Round 2
Reviewer 1 Report
Dear authors,
Thank for your work and patience in replying to my comments.
Of course, there is always room for more improvement
I would just refer:
- Tables 3 and 8 (split between pages);
- Figure 1 (with legend above the figure, when should be below).
- A large blank space in page 12.
Good work.
Author Response
Notes on the revision of the manuscript ID: sustainability-505455
Title
SUSTAINABILITY AND QUALITY MANAGEMENT IN THE LUXURY FURNITURE BUSINESS: A CIRCULAR ECONOMY PERSPECTIVE
Dear Editor and Reviewers,
We would like to thank you again for the opportunity to revise and resubmit our paper to Sustainability. We are glad that you appreciated the topic and our effort to follow your precious suggestions.
Below, we will answer point-by-point how we addressed the Reviewers’ comments.
Thank you againg for providing detailed and challenging reviews that have helped in improving the overall quality of the paper.
Responses to comments of Reviewer 1
Comment 1:
Dear authors,
Thank for your work and patience in replying to my comments.
Response 1:
Dear Reviewer,
Thank you for having appreciated our effort to follow all your precious comments in detail. We think that your comments have been of great utility to improve the overall quality of the paper.
Comment 2:
Of course, there is always room for more improvement
I would just refer:
- Tables 3 and 8 (split between pages);
- Figure 1 (with legend above the figure, when should be below).
- A large blank space in page 12.
Response 2:
Thank you for these clarifications of form. As suggested we have split Table 3 and 8 between pages, we have put the legend of Figure 1 below the figure and we have removed the large blank space in page 12. We hope that you could see these arrangements, and that these will not be moved by formatting when the system transforms our work from the word format to the pdf format. In any case, in the editing phase we hope that all the problems created by formatting will be eliminated. Thank you again for all your suggestions.
Reviewer 2 Report
After having read the revised paper and authors' responses to my first-round review, I can recommend this paper for publication.
Author Response
Notes on the revision of the manuscript ID: sustainability-505455
Title
SUSTAINABILITY AND QUALITY MANAGEMENT IN THE LUXURY FURNITURE BUSINESS: A CIRCULAR ECONOMY PERSPECTIVE
Dear Editor and Reviewers,
We would like to thank you again for the opportunity to revise and resubmit our paper to Sustainability. We are glad that you appreciated the topic and our effort to follow your precious suggestions.
Below, we will answer point-by-point how we addressed the Reviewers’ comments.
Thank you againg for providing detailed and challenging reviews that have helped in improving the overall quality of the paper.
Response to comments of Reviewer 2
Comment 1:
After having read the revised paper and authors' responses to my first-round review, I can recommend this paper for publication.
Response 1:
Dear Reviewer,
Thank you a lot for your help in reviewing this paper. We are very glad that you appreciated our effort to revise this paper following your precious comments and that you would recommend it for publication. We are grateful for all your effort in improving this paper.